# The *Arabidopsis HY2* Gene Acts as a Positive Regulator of NaCl Signaling during Seed Germination

**DOI:** 10.3390/ijms22169009

**Published:** 2021-08-20

**Authors:** Mingxin Piao, Jinpeng Zou, Zhifang Li, Junchuan Zhang, Liang Yang, Nan Yao, Yuhong Li, Yaxing Li, Haohao Tang, Li Zhang, Deguang Yang, Zhenming Yang, Xinglin Du, Zecheng Zuo

**Affiliations:** 1Jilin Province Engineering Laboratory of Plant Genetic Improvement, College of Plant Science, Jilin University, Changchun 130062, China; piaomingxin@163.com (M.P.); jczhang_gray@163.com (J.Z.); 13311577331@163.com (L.Y.); zhang_li18@mails.jlu.edu.cn (L.Z.); zmyang@jlu.edu.cn (Z.Y.); 2Basic Forestry and Proteomics Research Center, Fujian Agriculture and Forestry University, Fuzhou 350002, China; 17706314607@163.com (J.Z.); li18834414500@163.com (Z.L.); 18844199632@163.com (N.Y.); lyh317@163.com (Y.L.); fafu_lyx@163.com (Y.L.); tanghaohao0987@163.com (H.T.); 3College of Agriculture, Northeast Agricultural University, Harbin 150030, China; deguangyang@sina.com

**Keywords:** *Arabidopsis*, *HY2*, salt stress, seed germination, proteome, DRPs

## Abstract

Phytochromobilin (PΦB) participates in the regulation of plant growth and development as an important synthetase of photoreceptor phytochromes (phy). In addition, *Arabidopsis* long hypocotyl 2 (HY2) appropriately works as a key PΦB synthetase. However, whether *HY2* takes part in the plant stress response signal network remains unknown. Here, we described the function of HY2 in NaCl signaling. The *hy2* mutant was NaCl-insensitive, whereas HY2-overexpressing lines showed NaCl-hypersensitive phenotypes during seed germination. The exogenous NaCl induced the transcription and the protein level of *HY2*, which positively mediated the expression of downstream stress-related genes of *RD29A*, *RD29B*, and *DREB2A*. Further quantitative proteomics showed the patterns of 7391 proteins under salt stress. HY2 was then found to specifically mediate 215 differentially regulated proteins (DRPs), which, according to GO enrichment analysis, were mainly involved in ion homeostasis, flavonoid biosynthetic and metabolic pathways, hormone response (SA, JA, ABA, ethylene), the reactive oxygen species (ROS) metabolic pathway, photosynthesis, and detoxification pathways to respond to salt stress. More importantly, ANNAT1–ANNAT2–ANNAT3–ANNAT4 and GSTU19–GSTF10–RPL5A–RPL5B–AT2G32060, two protein interaction networks specifically regulated by HY2, jointly participated in the salt stress response. These results direct the pathway of *HY2* participating in salt stress, and provide new insights for the plant to resist salt stress.

## 1. Introduction

Saline soil is an unfavorable environmental factor that seriously affects seed germination, seedling growth, and even final yield in crops [1,2,3]. About 7% of the total land surface and 20% of the irrigated land are affected by soil with excessive salt concentration, and the situation is getting worse [4,5]. Global climate change and poor irrigation water quality are the main factors leading to soil salinization [6,7]. Therefore, it is urgent to study the molecular mechanism of plants’ adaption to saline soil and to enhance such adaptability of plants to saline soil through molecular genetic improvement.

As salt stress gives rise to ion stress, osmotic stress, secondary stress, and oxidative stress [8,9], it is crucial for plants to maintain the balance between ion, osmosis, and ROS. In addition, plants have evolved a series of mechanisms to maintain salt balance in the long process of evolution [10]. In terms of ionic stress, after the perception of a salt-stress signal induced by a high concentration of salt in plants, the salt receptor glycosyl inositol phosphoryl ceramide (GIPC) [11,12] can directly bind to the external Na^+^ to form a direct interaction, which activates the Ca^2+^ channel. The influx of Ca^2+^ thus drives the adaptive response to high salt levels, which promotes EF-hand Ca^2+^-binding proteins SOS3 to activate serine/threonine protein kinase SOS2, and then to activate Na^+^/H^+^ antiporter SOS1 to pump Na^+^ out of the cell, maintaining the salinity balance in vivo [13,14]. In terms of osmotic stress, the synthesis of compatible osmolytes is crucial for the maintenance of osmotic potential and protein structure in cells, including the expression of related genes as PM-located protein OSCA, MAPK cascades, SnRK2 isoforms, etc. [15,16,17], and the synthesis of the accumulation of related substances, such as proline, betaine, sugars, etc. [18,19,20]. In terms of oxidative stress, both the gene expressions, such as MAPKKK–MAPKK–MAPK (mitogen-activated protein kinase) cascade [21], and the ROS scavenging enzymes, such as superoxide dismutase (SOD), peroxidase (POD), catalase (CAT), etc. [22], act to maintain the balance of ROS in vivo. Therefore, improving the molecular mechanism of salt stress is very important for the strengthening of salt tolerance in plants.

PΦB is an open-chain tetrapyrrole chromophore, a critical synthetase for phytochromes to function as a light receptor to regulate plant growth and development [23,24]. *Arabidopsis*
*HY2* encodes a key synthase of PΦB [25,26], which is a ferredoxin-dependent biliverdin reductase that catalyzes the reduction in the A-ring 2,3,3^1^,3^2^-diene system to produce an ethylidene group for assembly with apophytochromes [27]. Furthermore, it has been reported that *HY2* induces the synthesis of *phyA* to inhibit the elongation of hypocotyl under the far-red light treatment [28]. Under the treatment of exogenous trehalose, the expression of *HY2* is upregulated by 2.8 times [29]. Besides, *HY2* participates in the apoplastic and chloroplastic redox signaling networks, being responsible for chlorophyll biosynthesis [30]. However, whether *HY2* is involved in the plant stress response signal network remains unknown.

In this study, we found that *Arabidopsis* PΦB synthetase *HY2* is a positive regulator in NaCl signaling during seed germination. Meanwhile, we conducted tandem mass tags (TMT)-based proteomics analysis [31] to compare col4 (wild type) with *hy2* mutant under salt stress to identify salt-stress-inducing HY2-specific responsive proteins. This would help to demonstrate the role of *HY2* in the salt stress response pathway. Our study actually reveals the plant salt stress response and identifies new elements in the salt stress pathway, which provides new insights into genetic engineering of the crops to improve salt tolerance and yield.

## 2. Results

### 2.1. Disruption of HY2 Reduces, and Overexpression of HY2 Enhances, NaCl Sensitivity during Seed Germination

To analyze the novel salt tolerance genes, we used a luciferase reporter system to construct different *Arabidopsis* transgene lines overexpressing firefly luciferase (LUC). Compared with col4 and *LUC-vector* lines, the lines expressing the PΦB synthetase HY2 showed a salt-hypersensitive phenotype and *hy2* mutant displayed a salt-resistant phenotype with 200 mM NaCl stress (Figure 1A). Gene lines overexpressing *HY2* and *hy2* mutant were then used to study the physiological function of *HY2* in seed germination (Appendix A). We found most *hy2* mutant seedlings germinated 3 days after being sown in the medium containing 200 mM NaCl, while the seedlings of col4 and *LUC-vector* needed 4–5 days to germinate; compared with col4, the germination rate of lines overexpressing *LUC–HY2* was significantly lower, but that of *hy2* mutant was obviously higher (Figure 1B).

The following QPCR analysis showed that, with 0 h and 1 h NaCl stress, the transcription level of *HY2* was undifferentiated; with 3 h and 5 h NaCl stress, the transcription level of *HY2* increased by 1.7 and 2.8 times, respectively (Figure 1C). The luciferase assay showed a 1.5-fold increase in the protein level of *HY2* after 3 h NaCl stress (Figure 1D). These results indicated that NaCl significantly mediates and upregulates the *HY2*, both on its transcription level and protein level. At the same time, we analyzed the expression of *HY2* in different tissues, showing that *HY2* was expressed in different tissues of *Arabidopsis*, but its expression level was the lowest in roots and the highest in flowers (Figure 1E).

### 2.2. Quantitative Proteomics Analysis of col4 and hy2 Mutant under SALT Stress

To identify the mechanism of *HY2* to the NaCl response pathway, we treated col4 and *hy2* mutant with NaCl stress, and then used TMT-based proteomics to figure out how HY2 protein specifically regulates the expression of salt-stress-related proteins at the protein level (Figure 2A, Appendix A). In our experiment, a total of 81,898 peptides and 68,002 unique peptides were matched with the *Arabidopsis* library; 9203 proteins were identified and 7391 proteins were quantified (Appendix A). The size of most identified proteins was in the range of 20–80 kDa, accounting for 74% of the identified proteins (Appendix A). The distribution of peptides indicated that the amount of the corresponding proteins decreased with the increase in the peptide number (Appendix A). The protein sequence coverage of 0–10%, 5–10%, 10–20%, 20–30%, 30–40%, 40–50%, 50–60%, and >60% was censused as 15%, 14.2%, 21.9%, 16.6%, 12.6%, 9.3%, 5.8%, and 4.6%, respectively (Appendix A). The principal component analysis (PCA) showed that the contribution ratio of principal component PC1 and PC2 was 54.0% and 21.3%, respectively, and the results showed an identical repeatability of the same experimental group. More interestingly, the expression level of proteins in col4 and *hy2* mutant significantly varied under NaCl stress, indicating that HY2 protein specifically regulates protein expression concerning the NaCl response pathway (Figure 2B and Appendix A). In addition, the heatmap of the expression level of all proteins showed a different protein expression pattern in col4 and *hy2* mutant (Appendix A). For a further verification of the repeatability of the experiment and an identification of the difference between col4 and *hy2* mutant on the protein expression level, we conducted Pearson correlation coefficient of the overall expression level, with a result that the correlation of repeatability was greater than 0.9, but the correlation of protein expression level in col4 and *hy2* mutant was about 0.5 (Figure 2C).

### 2.3. Function Analysis of Accurate Protein Quantification

Among the 7391 proteins accurately quantified, 5995 were quantified both in col4 and *hy2* mutant, 637 were specifically quantified in col4, and 759 in *hy2* mutant (Figure 3A, Appendix A). All of the quantified proteins were normalized with protein intensity of Actin1 and UBQ10. Proteins with a fold change ratio > 1.5 and *p* value < 0.05 were defined as DRPs, among which 31 (12 upregulated and 19 downregulated) specifically responded in col4 and 19 (19 downregulated) in *hy2* mutant (Figure 3B, Appendix A). It was obvious that the specific DRPs quantified in *hy2* mutant were far fewer than those quantified in col4, indicating a correspondence to the NaCl insensitivity of *hy2* mutant. For the function of these quantified proteins specifically regulated by HY2 under NaCl stress, we carried out GO enrichment analysis, the result of which indicated that the specific DRPs in col4 were the major players of ion homeostasis and flavonoid biosynthetic and metabolic processes to respond to salt stress (Figure 3C, Appendix A), while the specific DRPs in *hy2* mutant were mainly involved in hormone response pathways (JA, SA, ABA, and ethylene), cellular detoxification, and the ROS metabolic pathway to respond to salt stress (Figure 3D, Appendix A).

### 2.4. Function Analysis of CRPs

The 5995 quantified proteins identified in both col4 and *hy2* mutant presented different protein expression patterns (Figure 4A). Among these shared quantified proteins, 194 DRPs were quantified in col4 and 97 DRPs were quantified in *hy2* mutant (Appendix A)*,* the result of which showed that DRPs quantified in *hy2* mutant were significantly fewer than those in col4 (only about 50% of col4 responsive proteins), corresponding to the salt-resistant phenotype of *hy2* mutant again (Appendix A). Within those DRPs quantified in col4 and those in *hy2* mutant, we identified 63 shared DRPs (63 downregulated), 131 col4-specific DRPs (84 upregulated and 47 downregulated), and 34 *hy2*-specific DRPs (5 upregulated and 29 downregulated) (Figure 4B, Appendix A). We then classified these DRPs into two groups: one was the shared DRPs in col4 and *hy2* mutant, which responded to NaCl stress but were not specifically regulated by HY2; the other was col4-specific or *hy2*-specific DRPs, which were specifically regulated by HY2. We next conducted GO analysis to study the various functions of those proteins and found that the proteins that were regulated by NaCl stress but not specifically regulated by HY2 mainly took part in the salt stress response, stress and stimulus response, cell wall organization, ROS metabolic pathway, detoxification, and lipid transport (Appendix A, Appendix A). We also found that, responding to salt stress, the proteins specifically regulated by HY2 were mainly involved in oxidative stress response, ion transmembrane transport, photosynthesis, and response to water and detoxification in col4 (Figure 4C, Appendix A), while those in *hy2* mutant were involved in defense response, biotic stimulus response, proteolysis, and the response to wounding (Figure 4D, Appendix A).

### 2.5. HY2-Specific Regulated Pathway Relating to the Salt Stress Response

We investigated the function of the HY2-regulated pathway relating to the salt stress response. Mainly three groups of the pathway were classified: overcoming the salt stress, redox states, and hormones and metabolites response. As shown in Figure 5A, HY2 directly regulated the pathways of flavonoid biosynthetic, ion homeostasis, and ion transport relating to overcoming salt stress. Furthermore, HY2 affected the salt stress response though the signaling pathways of ROS metabolism, H_2_O_2_ metabolic, oxidative stress, abscisic acid biosynthetic, ethylene response, jasmonic acid response, etc., which belonged to the redox states group, respectively. We compared the population of redox-related proteins in col4 and *hy2* mutant. As shown in Figure 5B,C, except for the overlap redox-related protein, only two proteins belonged to col4. In contrast, there were 29 proteins that were only responsive in *hy2* mutant, which suggested that HY2 protein significantly affects the redox response in plants.

We further investigated the relationship between HY2 and Na^+^/K^+^ homeostasis, which play a key role in plants to overcome salinity stress. Since the regulation of Na^+^/K^+^ homeostasis was mainly involved in these biological processes (e.g., ion osmoregulation, inorganic ions regulation, organics regulation, and related enzyme activity regulation) [8,9], we analyzed the HY2-regulated pathway related to Na^+^/K^+^ homeostasis. As shown in Figure 6A–D, HY2 regulated the pathway involved in Na^+^/K^+^ homeostasis regulation, including ion osmoregulation, organics regulation, and related enzyme activity regulation. More importantly, we found that the expression of a few members of K^+^ transporter family ATANNs was decreased in *hy2* mutant compared to col4. As reported previously [32], the reduction in ATANNs expression would induce the salt-stress-insensitive phenotype, similar to the phenotype of *hy2* mutant. Furthermore, a few members of the PRX family related to organics regulation and elated enzyme activity regulation were also regulated by HY2. These results suggested that HY2 may regulate the expression of the factor related to ion osmoregulation, organics regulation, and enzyme activity regulation to affect the Na^+^/K^+^ homeostasis and salinity tolerance of the plant.

### 2.6. Disruption of HY2-Altered Expression of a Set of Stress-Responsive Genes

NaCl stress is one of the stress factors that affects the growth and development of seeds, and induces the expression of stress-related proteins, such as seed growth and development related proteins and ABA-pathway-related proteins. Therefore, we verified the expression patterns of proteins related to salt stress, seed growth and development, and ABA pathway in col4 and *hy2* mutant (Figure 7A), from which we found that 15 proteins were mediated by NaCl stress but not by HY2 in a specific way (gray dot), while 32 were specifically mediated by HY2 (green and yellow dot). All of the proteins were then divided into different groups: proteins involved in the salt stress, proteins participating in the ABA pathway, and proteins taking part in the growth and development of seeds, among which the number of proteins specifically regulated by HY2 (col4-specific and *hy2*-specific) were 22, 10, and 5, respectively, while the number of proteins specifically regulated by NaCl stress were only 9, 7, and 1 (less than half the number of proteins specifically regulated by HY2), respectively. These results showed, under NaCl stress conditions, that proteins involved in the salt stress pathway dominated, accounting for 66%, and those involved in the ABA pathway and seed growth/development took the secondary place (Figure 7B), indicating that HY2 simultaneously regulates the protein expression related to salt stress, ABA pathway, and seed growth and development. As *HY2* is a potential regulator involved in NaCl signaling (Figure 1A), the expression of stress inductive genes, such as *RD29A*, *RD29B*, and *DREB2A*, was tested in *hy2* mutant. We tested the inducible genes under the conditions of 0 h–5 h stress of 200 mM NaCl, and found that the expression levels of *RD29A* and *RD29B* were undifferentiated at 0 h and 1 h in *hy2* mutant, but significantly decreased at 3 h and 5 h when compared with those in col4 at the same conditions (Figure 7C,D); the expression level of *DREB2A* remained the same at 0 h, 1 h, and 5 h, and was significantly downregulated at 3 h only (Figure 7E), indicating that *HY2* induces the expression of NaCl inducible genes and positively regulates NaCl signaling.

### 2.7. Interaction Network of HY2-Specific DRPs

Protein interaction networks were generated to evaluate the interaction of the DRPs (known and unknown proteins) specifically regulated by HY2 (Figure 8). We selected 22 proteins specifically regulated by HY2 under salt stress, among which 17 were col4-specific (14 upregulated and 3 downregulated) and five were *hy2*-specific (five downregulated). The results showed that ANNAT1–ANNAT2–ANNAT3–ANNAT4, the family members of Annexins (a family of calcium dependent membrane binding proteins), were involved in salt-stress responses specifically regulated by HY2 in a mutual-functioning way. The score of association between ANNAT1 and ANNAT2 was 0.915, which was the highest, while that between ANNAT1 and ANNAT4 was 0.805, which was the lowest. Glutathione transferase GSTU19 induced by drought, oxidative stress, and hormonal responses; Glutathione S-transferase PHI 10 (GSTF10) involved in water deprivation and toxin catabolic process; 60S ribosomal protein L5 (RPL5A, RPL5B) responsible for the synthesis of proteins in the cell as the component of the ribosome; and ribosomal proteins of L7Ae/L30e/S12e/Gadd45 family protein (AT2G32060) involved in translation all interactionally participated in salt-stress responses specifically regulated by HY2. The association score between AT2G32060 and RPL5A/RPL5B was 0.999, while that between GSTF10 and RPL5A/RPL5B was 0.596. We, therefore, identified two interaction protein networks specifically regulated by HY2 under salt stress.

## 3. Discussion

As an increasingly serious abiotic stress factor, salt stress induced by saline soil threatens the growth and development of plants. Therefore, it is of great importance to explore related regulatory factors of salt stress and improve related pathways of salt stress [1,2,3]. In the current study, we found that *Arabidopsis*
*HY2* regulates seed germination and seedling growth under salt stress as PΦB synthase [25,26]. We also found that *HY2* is a positive regulator to regulate the expression of downstream related genes with the physiological phenotype and biochemical analysis, and identified key proteins specifically regulated by HY2 and correlated pathways with proteomics.

Phytochrome is an important photoreceptor for plants to sense environmental changes [33,34,35]. *Arabidopsis hy1* and *hy2* mutants cannot synthesize photoactivated photoactive phytochrome due to the lack of PΦB biosynthesis, resulting in impaired photomorphogenesis [26,28]. In addition, *HY2* participates in the regulation of hypocotyl elongation under far-red light, the induction of exogenous trehalose, and the biosynthesis of chlorophyll [28,29,30], but whether it is involved in the stress response pathway is unknown. Previous studies have shown that 150–250 mM NaCl stress has serious effects on plant growth and development [36,37,38]; therefore, we used 200 mM NaCl stress as the salt stress screening condition. We next used the luciferase reporting system [39,40] to obtain a large number of LUC tag, in which *HY2* a positive regulator of salt stress, was screened out. The phenotype of *HY2* was then observed and the germination rate under 3–5 d NaCl stress was recorded, as a result of which we found that the best time to observe the phenotype is after 4 d treatment of NaCl stress. The 3 d treatment and 5 d treatment is either too early or too late for such observation.

To identify the proteins and pathways specifically regulated by HY2 under salt stress, we conducted a TMT6-labeled proteomics analysis and accurately quantified the changing patterns of 7931 proteins under salt stress. Interestingly, we found that proteins specifically regulated by HY2 could mediate the flavonoid biosynthetic process (RNS1, UGT72E1), hormone response pathway (NHL6, HR4, GAMMA-VPE), and photosynthesis pathway (PSI-P, GUN5, LHB1B1, CAB1, Lhca6, LHCA1, ATPC1, NdhL, PPC1, DUF1995 and PPC2), apart from their regulation on ion homeostasis, cellular detoxification, and reactive oxygen species metabolic pathway. According to previous studies, with salt stress, the expression patterns of 214 flavonoid biosynthetic genes in soybean change [41], the content of flavonoid in *Solanumnigrum* raises with the increase in salt concentration [42], and 584 genes in *Elaeagnusangustifolia* L. are identified and involved in photosynthetic pathways [43]. Besides, a variety of phytohormones, including ABA, GA, auxin, and CK, are involved in the regulation of salt stress response in plants [44]. These results suggest that *HY2* may be involved in the interaction between the salt stress pathway and many other pathways, such as the flavonoid pathway, plant hormone pathway, and photosynthesis pathway.

In this study, we found that HY2 specifically regulated pathways, such as ion balance, ion transmembrane transport, ROS metabolism, antioxidant response, and photosynthesis, etc. Numerous previous studies have shown that salt stress is closely related to antioxidant indexes in plants, such as the content of H_2_O_2_, NADH, and ATP, as well as Na^+^/K^+^ ion balance [45]. Besides, studies have shown that HY2 participates in apoplastic and chloroplastic redox signaling networks, being responsible for chlorophyll biosynthesis [30]. In this study, we found that HY2 regulated the ion osmoregulation, organics regulation, and related enzyme activity regulation pathways, which were involved in Na^+^/K^+^ homeostasis regulation. Furthermore, a few members of K^+^ transporter family ATANNs and PRX family were regulated by HY2 during salt stress, which directly affected the ion osmoregulation, organics regulation, and related enzyme activity regulation to maintain Na^+^/K^+^ homeostasis in salt stress. Therefore, we speculated that, under salt stress, HY2 would be induced to upregulate the expression, leading to the imbalance of reactive oxygen species and Na^+^/K^+^ in vivo and, thus, the damage of plants by reactive oxygen species and ion stress, the decrease in chlorophyll content, and the serious impact on photosynthesis.

Earlier studies have shown that the expression levels of important downstream genes of *RD29A*, *RD29B*, and *DREB2A* are induced by various abiotic stresses, including drought, chilling stress, and salt stress [46,47,48,49]. In this study, the expression levels of downstream genes were all induced to upregulate under 0–5 h salt stress, which is consistent with previous studies. Meanwhile, in *hy2* mutant, the upregulated expression of downstream genes was inhibited, corresponding to the role of *HY2* as a positive regulator of salt stress. We used interaction network analysis and identified two interaction networks specifically regulated by HY2 under salt stress, which are ANNAT1–ANNAT2–ANNAT3–ANNAT4 and GSTU19–GSTF10–RPL5A–RPL5B–AT2G32060. Previous studies have shown that ANNAT1–ANNAT2–ANNAT3–ANNAT4, as an important membrane-binding protein, participates in drought, salt stress, chilling stress, and other abiotic stresses [32,50,51,52]; RPL5A–RPL5B, a 60S ribosomal protein, is involved in cold and water-deficit stresses [53,54]. These two candidate salt stress networks provide an important theoretical basis for the study of HY2′s participation in salt stress, but its specific mechanism needs further experimental verification. The means by which *HY2* gene accurately acts on the salt stress response pathway is still an open question.

## 4. Materials and Methods

### 4.1. Plasmid Construction

Plasmids used in this study were generated by In-Fusion cloning [55,56] (https://www.takarabio.com/products/cloning/in-fusion-cloning (accessed on 25 July 2020)). PEGAD–LUC vectors were used to create overexpression transgenic lines. The LUC fragment was cloned into PEGAD-MYC [57,58] to generate PEGAD–LUC vectors. The coding sequences (CDS) of *HY2* were amplified from *Arabidopsis* cDNA made previously by PCR, and the purified PCR products were then subcloned into Ecor I/Hind III-digested PEGAD–LUC vectors through In-fusion cloning.

### 4.2. Plant Materials and Growth Conditions

The wild-type plant used in this study was *Arabidopsis* col4. T-DNA insertion line *hy2* mutant (SALK_104923C) was obtained from *Arabidopsis* Biological Resource Center (https://abrc.osu.edu/ (accessed on 4 June 2018)) and identified by the *HY2*-specific primers and T-DNA left-board primers. Transgenic *Arabidopsis* expressing the LUC fusion protein (LUC–HY2) was prepared by floral dipping method [59] in col4 background. LUC positive lines were screened with glufosinate, CCD camera, and Western blot with Anti-luciferase antibody (Sigma, L2164, 3050 Spruce Street, Saint Louis, MO 63103, USA). Col4, *LUC-vector*, *hy2* and *LUC–HY2* were grown in Petri dishes in half-strength Murashige and Skoog salts (1/2 MS; Sigma), 1% (*w*/*v*) sucrose (Sigma), and 0.8% (*w*/*v*) agar (Sigma) in continual illumination (120 μmol m^−2^ s^−1^) at 22 °C, unless specifically indicated [13]. Seeds were sown on 1/2 MS media, placed at 4 °C for 3 days in the dark, and then transferred to growth rooms. The primers used for genotyping *HY2*-overexpressing lines and *hy2* mutant are listed in Appendix A.

### 4.3. Salt Sensitivity Assay

Sterilized seeds were sown on 1/2 MS medium (as mock) or 1/2 MS medium containing indicated concentrations of 200 mM NaCl at pH 5.8 with 0.8% (*w*/*v*) agar. After 3–5 d, seedlings were photographed, and the germination rate was determined as a percentage of the total number of seeds plated. Germination was defined as an obvious emergence of the radicle through the seed coat [60]. At least 64 seedlings were observed per line, and each experiment was repeated three times.

### 4.4. LUC Activity Recordings

*LUC–HY2* overexpression lines were sown in a 96-well plate containing 1/2 MS supplemented with 0.2% (*w*/*v*) sugar, 0.4% (*w*/*v*) agar, 200 mM NaCl, and 1 mM D-luciferin (potassium salt), with 10 seeds per well for each individual line. Seedlings were transferred to darkness for LUC activity detection [61]. LUC signals were detected every 10 min, with a detecting period of 5 h.

### 4.5. QPCR Assay

The total RNA of seedlings was extracted with RNeasy Plant Mini kit (QIAGEN, 74904, Germany). The total RNA (2 μg) was treated with DNase I (Takara, Beijing, China) to eliminate genomic DNA contamination. Then, the cDNA was synthesized by SuperScript II Reverse Transcriptase (Invitrogen) using radom hexamer primers (Promega), then performed with 1 µL of template cDNA, 1 µL of forward primer (0.2 µm), 1 µL of reverse primer (0.2 µm), and 10 µL TB Green Premix Ex Taq, in a total reaction volume of 20 µL, successively. qRT-PCR was eventually carried out to a Mx3005P Real-Time PCR System [62,63]. The qPCR signals were normalized to that of the reference gene *Actin1* using the ΔCT method [64]. There were three replicates in each sample. The primers are listed in Appendix A.

### 4.6. Tandem Mass Tags (TMT)-Based Proteomics Analysis

In order to compare the proteomics of col4 and *hy2* mutant, 7-day-old seedlings were treated with 200 mM NaCl or water (as mock) for 5 h. Approximately 0.5 g of seedlings were extracted with protein lysis buffer (10 mM Tris-HCl, pH 8.0, 5 mM EDTA, 1% SDS, 8 M urea, 20 mM dithiothreitol, 1× EDTA-free protease inhibitor cocktail tablets). Protein was digested using a filter-aided sample preparation (FASP) method [65]. Digested peptides were dried using a CentriVap (Thermo Fisher, Shanghai, China) and pre-fractionated with Ultimate 3000 (Thermo Fisher scientific, Waltham, MA, USA) [66]. Peptide was analyzed by on-line nanospray LC-MS/MS on an Orbitrap Fusion coupled to an EASY-nano-LC system (Thermo Scientific, MA, USA) [67,68]. All MS/MS raw data were analyzed using Proteome Discoverer 2.1 (Thermo Fisher Scientific, San Jose, CA, USA; version 2.1) [69] and Scaffold Q+ (version Scaffold_4.7.1, Proteome Software Inc., Portland, OR, USA) [70].

### 4.7. GO Enrichment Analysis

As per GO vocabulary, the sequences were characterized by OMICSBOX (www.biobam.com/omicsbox (accessed on 18 May 2021)) to predict the role of contigs in biological functions (BP, MF, and CC). GO enrichment analysis of DRPs was carried out to determine their roles in BP, MF, and CC through OMICSBOX [71].

### 4.8. Interaction Network Analysis

Protein–protein interaction (PPI) networks of HY2 specifically regulated proteins under salt stress were built using STRING v11, with a confidence score threshold of 0.9 (https://www.string-db.org/ (accessed on 10 June 2021)) [72,73].

### 4.9. Quantification and Statistical Analysis

All statistical data were collected in a GraphPad Prism 8.0.2. ANOVA with two-tailed Student’s *t*-test [74] was used to evaluate statistical significance, while ^ns^
*p* > 0.05, * *p* < 0.05, ** *p* < 0.01, *** *p* < 0.001. All data were reported as mean ± SD.

## Figures and Tables

**Figure 1 ijms-22-09009-f001:**
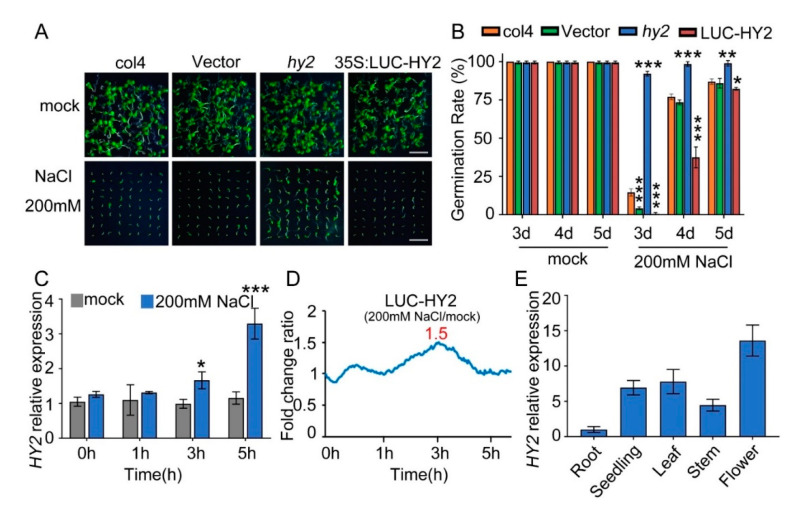
NaCl sensitivity of *hy2* mutant and *HY2*-overexpressing lines. (**A**) Phenotypic comparison. Col4, *LUC*-*vector*, *hy2* mutant, and *LUC–HY2* overexpression lines were sown, respectively, on 1/2 MS medium (as mock) or 1/2 MS medium containing indicated concentrations of 200 mM NaCl for 4 d. Scare bar: 1 cm. (**B**) Seed germination assay. Seeds were transferred to 1/2 MS containing 200 mM NaCl, and then the seed germination rate was calculated at 3–5 d. Data are shown as mean ± SD (*n* = 3). More than 64 seeds were measured in each replicate. (**C**) QPCR analysis of *HY2* expression in 5-day-old col4 seedlings treated with or without 200 mM NaCl for 0–5 h. Data are shown as mean ± SD (*n* = 3). (**D**) LUC signals in 5-day-old *LUC–HY2* overexpression line seedlings treated with or without 200 mM NaCl. Signals were detected every 10 min, and the detecting period was 5 h. (**E**) QPCR analysis of *HY2* expression in different tissues of *Arabidopsis*. Data are shown as mean ± SD (*n* = 3). Asterisks in (**B**,**C**) indicate statistically significant differences compared with relevant col4 plants: * *p* < 0.05, ** *p* < 0.01, *** *p* < 0.001.

**Figure 2 ijms-22-09009-f002:**
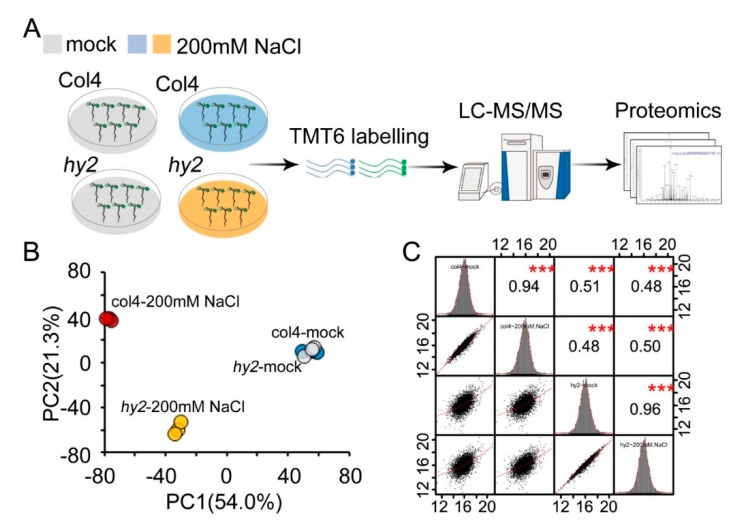
Quantitative proteomics analysis of col4 and *hy2* mutant under salt stress. (**A**) Workflow of proteomics analysis. 5-day-old col4 and *hy2* mutant seedlings were treated with or without 200 mM NaCl for 5 h. The proteomics analysis consists of three steps. Step 1, proteins were extracted from tissues and proteolytically digested. Step 2, TMT6 labelling. Step 3, nano-HPLC-MS/MS. (**B**) Unsupervised principal component analysis (PCA) of quantitative proteomics data. (**C**) Scatterplot matrices by which linear and nonlinear relations were investigated. The value represents the Pearson correlation between treatments. The red asterisk showed in (**C**) represents a significant difference between treatments: *** *p* < 0.001.

**Figure 3 ijms-22-09009-f003:**
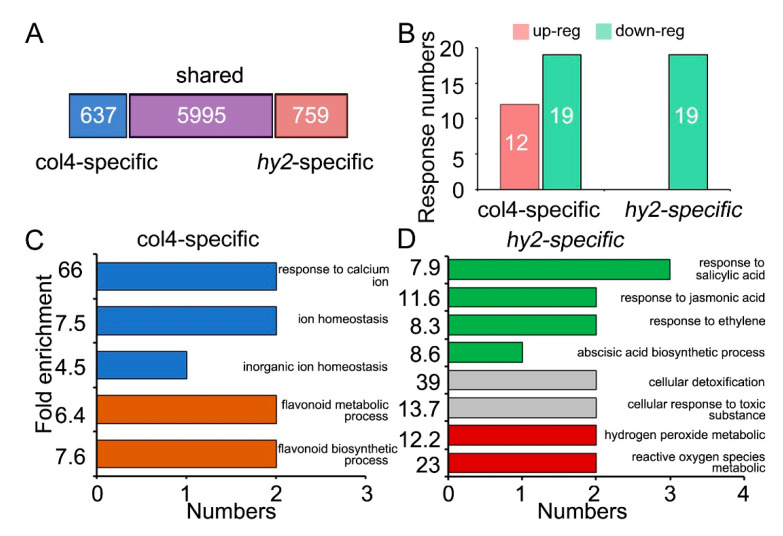
Function analysis of specific quantified proteins. (**A**) The number of specific and shared quantified proteins between col4 and *hy2* mutant under salt stress. (**B**) The response number of col4-specific and *hy2*-specific DRPs under salt stress. The red and green squares represent upregulated and downregulated proteins, respectively. (**C**) The GO enrichment analysis of col4-specific quantified and response proteins. (**D**) The GO enrichment analysis of *hy2*-specific quantified and response proteins.

**Figure 4 ijms-22-09009-f004:**
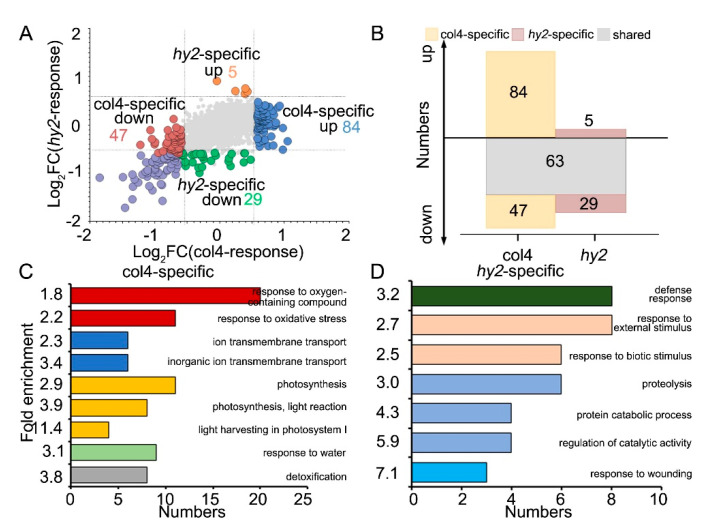
Function analysis of DRPs. (**A**) Protein expression pattern of col4 and *hy2* shared quantified proteins. The abscissa and ordinate represent the protein pattern of col4 and *hy2* mutant, respectively. Color denotes proteins regulated similarly by col4 and *hy2* (grey), or specifically by col4 (blue and red) or *hy2* (orange and green). (**B**) The number of distinct proteins significantly up- or downregulating col4 and *hy2* mutant. Color denotes proteins regulated similarly by col4 and *hy2* (grey), or specifically by col4 (orange) or *hy2* (brown). (**C**) The GO enrichment analysis of col4-specific response proteins. (**D**) The GO enrichment analysis of *hy2* specific response proteins.

**Figure 5 ijms-22-09009-f005:**
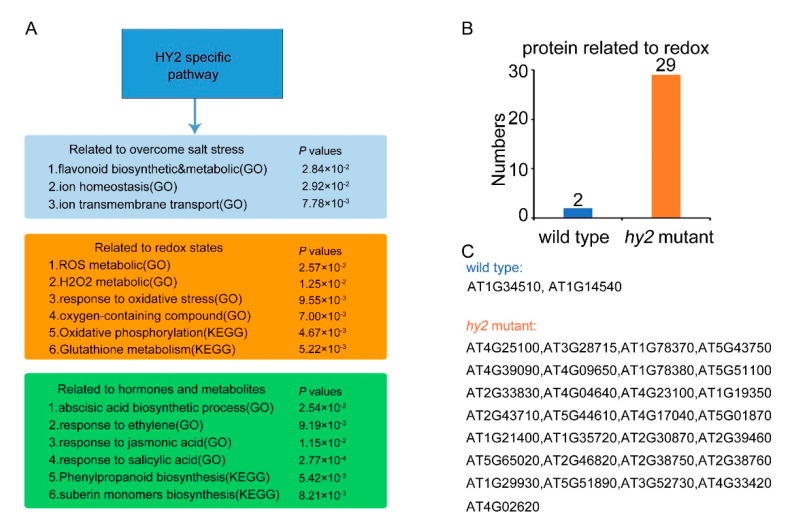
HY2-specific regulated pathway analysis. (**A**) GO enrichment and KEGG enrichment analysis models of HY2 specifically regulated proteins under salt stress. The light blue module represents the pathway related to the overcoming of salt stress, the yellow module represents the pathway related to redox states, and the green module represents the pathway related to hormones and metabolites. (**B**,**C**) The proportion of redox-related proteins in wild-type and *hy2* mutant among the proteins specifically regulated by HY2.

**Figure 6 ijms-22-09009-f006:**
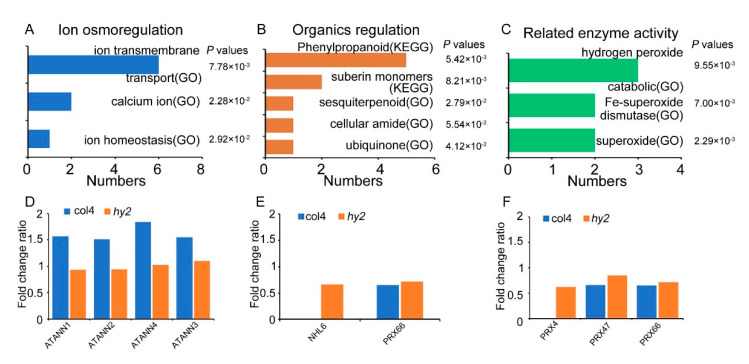
The important HY2-regulated salt stress pathways and the changing patterns of their key proteins in col4 and *hy2* mutant. (**A**,**D**) Ion-osmoregulation-related pathways and the changing patterns of the key proteins ATANN1, ATANN2, ATANN3, and ATANN4. (**B**,**E**) Pathways related to organics regulation and the changing patterns of the key proteins NHL6 and PRX66. (**C**,**F**) Pathways related to enzyme activity of free-radical-scavenging system and the changing patterns of the key proteins PRX4, PRX47, and PRX66.

**Figure 7 ijms-22-09009-f007:**
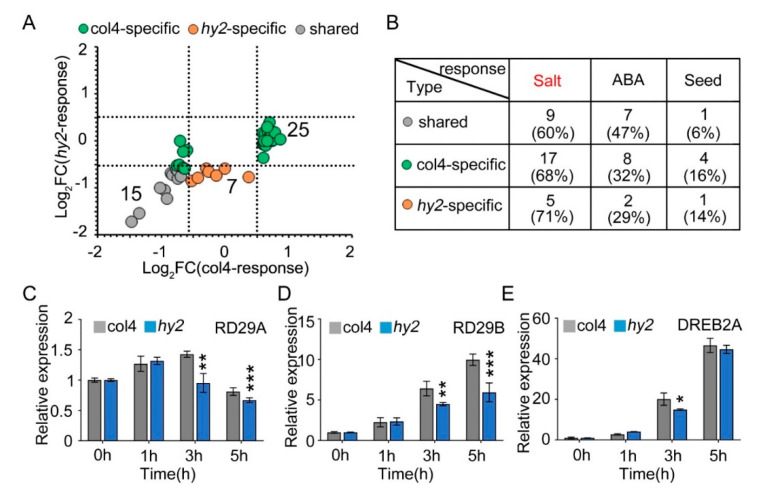
The expression of stress-responsive genes. (**A**) Protein expression pattern of col4 and *hy2* shared proteins involved in salt, ABA, and seed-related pathway. Color denotes proteins regulated similarly by col4 and *hy2* (grey), or specifically by col4 (green) or *hy2* (orange). (**B**) The number and proportion of col4 and *hy2* shared proteins involved in salt, ABA, and seed-related pathway. (**C**–**E**) The expression of *RD29A*, *RD29B*, and *DREB2A* in col4 and *hy2* mutant seedlings treated with exogenous NaCl. The 5-day-old col4 and *hy2* mutant seedlings were transferred to 1/2 MS solution with or without 200 mM NaCl for 0–5 h, and then the seedlings were harvested for QPCR. Data are shown as mean ± SD (*n* = 3). Asterisks showed in (**B**–**D**) indicate statistically significant differences compared with relevant col4 plants: * *p* < 0.05, ** *p* < 0.01, *** *p* < 0.001.

**Figure 8 ijms-22-09009-f008:**
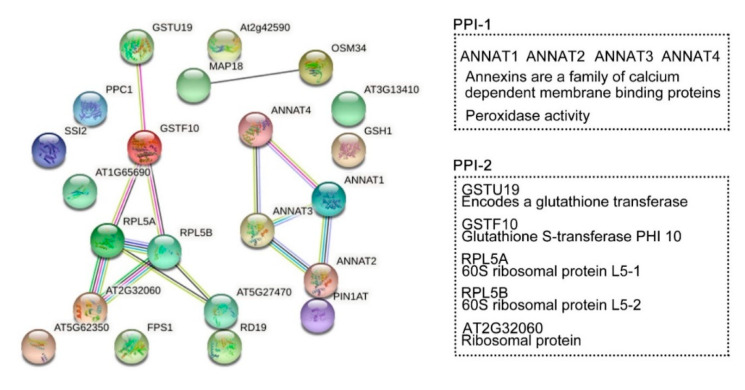
Protein-protein interaction (PPI) networks. STRING analysis using the HY2-specific regulated proteins under salt stress. Network nodes represent proteins. Edges represent protein–protein associations. Known interactions: 
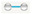
 from curated databases; 
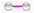
 experimentally determined. Predicted interactions: 
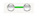
 protein neighborhood; 
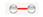
 protein fusions; 
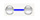
 protein co-occurrence. Others: 
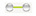
 textmining; 
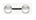
 co-expression; 
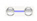
 protein homology. Tow PPI specifically regulated by HY2. PPI-1: ANNAT1–ANNAT2–ANNAT3–ANNAT4. PPI-2: GSTU19–GSTF10–RPL5A–RPL5B–AT2G32060.

## Data Availability

The data that supports the findings of this study are available in the Appendix A of this article. The mass spectrometry proteomics data have been deposited to the ProteomeXchange Consortium via the PRIDE [75] partner repository with the dataset identifier PXD027204.

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
