# Peer review of "The Arabidopsis HY2 Gene Acts as a Positive Regulator of NaCl Signaling during Seed Germination"

_ijms, 2021, doi:10.3390/ijms22169009_

Round 1

Reviewer 1 Report

The ms “The Arabidopsis HY2 gene acts as a positive regulator of NaCl signaling during seed germination” is a nice study that investigates the role of HY3 gene in salt stress signalling in A. thaliana. The authors well described the main results, which sound with the initial hypothesis, but I think the discussion should be improved, i.e., not limited to the comparison with the current literature or repeating the results. The results are also briefly presented at the end of the introduction, but I think the authors should not anticipate them otherwise the text becomes very redundant. With respect to the figure, if possible, I suggest avoiding the gene functions to overlap with the bars in the graphs.

The manuscript needs moderate English (and typing) revision.

Author Response

Response to Reviewer 1 Comments

   We thank the reviewer for your interest in our research, and the great efforts in helping improving the quality of our manuscript. We have replied the questions point-by-point as following:

Point 1: The work of the authors was performed at a high level. Due to the careful statistical processing of the results, the data are not in doubt. My only question is about setting up the experiment. Why did the authors choose only one concentration of sodium chloride? The choice of this concentration should be more carefully justified. It is necessary to show how seed germination occurs at other concentrations on the growth curve, and to justify the choice of this point.

Response 1: We thank the reviewer for the interest in our research and the suggestions. "why choose only one concentration of sodium chloride? The choice of this concentration should be more carefully justified ". Yes, we have carefully justified the NaCl concentration for phenotype analyses, which haven’t presented the details in the method.

    When we designed this experiment, we shared the similar idea with the reviewer. We carefully chose the threshold of concentration of sodium chloride (150-250 mM) for the phenotype screening of seedlings confronting with salt stress, based on the previous studies[1-3]. So, at the beginning of this study, we used three different NaCl concentrations (150, 200 and 250 mM) to examine the phenotype of our candidate library respectively.

   After we identified the salt stress phenotype of HY2 with different NaCl concentrations (150, 200 and 250 mM), we accessed the optimum NaCl concentrations for HY2 mutant phenotype. It exhibited that the sensitive phenotype of HY2 overexpressed plants was not significant at the concentration of 150 mM; the insensitive phenotype of mutant plants (hy2 mutant) was not obvious at the concentration of 250 mM. However, we found the HY2 overexpressing lines and hy2 mutant presented the optimum phenotype (opposite phenotype for salt stress) at the concentration of 200 mM NaCl, compared with wild type col4. Based on the above conditions, we selected NaCl with 200 mM concentration as the final condition for phenotypic presentation for this study.

  1. Ma, L.; Zhang, H.; Sun, L.; Jiao, Y.; Zhang, G.; Miao, C.; Hao, F. NADPH oxidase AtrbohD and AtrbohF function in ROS-dependent regulation of Na⁺/K⁺homeostasis in Arabidopsis under salt stress. Journal of experimental botany 2012, 63, 305-317.
  2. Du, Y.T.; Zhao, M.J.; Wang, C.T.; Gao, Y.; Wang, Y.X.; Liu, Y.W.; Chen, M.; Chen, J.; Zhou, Y.B.; Xu, Z.S.; et al. Identification and characterization of GmMYB118 responses to drought and salt stress. BMC plant biology 2018, 18, 320.
  3. Ma, L.; Ye, J.; Yang, Y.; Lin, H.; Yue, L.; Luo, J.; Long, Y.; Fu, H.; Liu, X.; Zhang, Y.; et al. The SOS2-SCaBP8 Complex Generates and Fine-Tunes an AtANN4-Dependent Calcium Signature under Salt Stress. Developmental cell 2019, 48, 697-709.e695.

Reviewer 2 Report

The work of the authors was performed at a high level. Due to the careful statistical processing of the results, the data are not in doubt. My only question is about setting up the experiment. Why did the authors choose only one concentration of sodium chloride? The choice of this concentration should be more carefully justified. It is necessary to show how seed germination occurs at other concentrations on the growth curve, and to justify the choice of this point.

Author Response

Response to Reviewer 2 Comments

    We thank the reviewer for your interest in our research, and the great efforts in helping improving the quality of our manuscript. We have replied the questions point-by-point as following:

Point 1: First of all, as it is well known from several studies that HY2 participates in the apoplastic and chloroplastic redox signaling networks, being responsible for chlorophyll, do authors have the answer in which pathway HY2 participates to overcome salt stress either which redox states, which hormones and metabolites?

Response 1: This is an interesting question. To address this issue, we re-analyzed the HY2 specifically regulated pathway from our database. According to the question raised by the reviewer, we classified those HY2-regulated pathways into four groups: (1) Related to overcoming the salt stress, (2) Related to redox states, (3) Related to hormones and metabolites and (4) Others. As shown in the newly added Figure 5A we found HY2 not only regulated the pathway of flavonoid biosynthetic, ion homeostasis and ion transport relating to overcoming the salt stress directly, but affected the signaling pathway of ROS metabolism, H2O2 metabolic, oxidative stress, etc., which were related to redox states and the signaling response of hormones and metabolites to influence the salt stress response.

    We thank the reviewer for raising this question and have added the new figures and sentences in the revised manuscript to explain this issue: “We further investigated the function of HY2 regulated pathway relating to the salt stress response. Mainly three groups of the pathway were classified: overcoming the salt stress, redox states and hormones & metabolites response. As shown in Figure 5A, HY2 directly regulated those pathways of flavonoid biosynthetic, ion homeostasis and ion transport relating to overcome salt stress. Furthermore, HY2 affected the salt stress response though those signaling pathways of ROS metabolism, H2O2 metabolic, oxidative stress, abscisic acid biosynthetic, ethylene response, jasmonic acid response, etc., which belonged to the redox states group, respectively.”

Point 2: If the Authors have any information what would be the redox amount in wild type and as well as hy2 mutant?

Response 2: Here we thank the reviewer again for his question 1. Since we have re-analyzed our data for question 1, we could statistically analyze the gene population related to redox in wild type and hy2 mutant, which reflected the discrepancy of redox amount in wild type and hy2 mutant. As shown in the newly added Figure 5B, 5C, except the overlap redox-related protein, only 2 proteins belonged to wild type. In contrast, there were 29 proteins only responsive in hy2 mutant, which suggested that HY2 protein significantly affects the redox response in plant.

Point 3: However, the maintaining a balanced of cytosolic Na+/K+ ratio is another key to the salinity tolerance mechanism, so this information is lack to the manuscript either the transporters of Na+/Kor total Na+/Kcontent in the whole seedlings. Is it possible to test the transporter's expression of Na+/K+ under salt stress in wild-type and hy2 mutants? The data are not really enough to support the Journal.

Response 3: We thank the reviewer for pointing out this relevant issue that we had previously overlooked. We understood that the maintenance of Na+/K+ homeostasis is important for plant survival during salt stress. The regulation of Na+/K+ homeostasis was involved in several processes, e.g. ion osmoregulation, inorganic ions regulation, organics regulation and related enzyme activity regulation[1,2].

    In this revision, we re-analyzed our database to investigate the relationship between HY2 and Na+/K+ homeostasis. We also added the new figure and several sentences to explain the relationship between HY2 and Na+/K+ homeostasis: “We further investigated the relationship between HY2 and Na+/K+ homeostasis which play a key role in plant to overcome the salinity stress. Since The regulation of Na+/K+ homeostasis was mainly involved in these biological processes (e.g. ion osmoregulation, inorganic ions regulation, organics regulation and related enzyme activity regulation) [1,2], we analyzed the HY2-regulated pathway related to Na+/K+ homeostasis. As shown in the newly added Figure 6A-6D, HY2 regulated the pathway being involved in Na+/K+ homeostasis regulation, including: ion osmoregulation, organics regulation and related enzyme activity regulation. More importantly, we found that the expression of a few members of K+ transporter family ATANNs was degreased in hy2 mutant, compared to wild type. As reported previously[3], the reduction of ATANNs expression would induce the salt-stress-insensitive phenotype, similar to the phenotype of hy2 mutant. Furthermore, a few members of PRXs family related to organics regulation and elated enzyme activity regulation were also regulated by HY2. These result suggested that HY2 may regulate the expression of the factor related to ion osmoregulation, organics regulation and enzyme activity regulation to affect the Na+/K+ homeostasis and salinity tolerance of plant.”

Point 4: Please correct how many reference genes are used to normalize the target genes.

Response 4: We thank the reviewer for pointing out this relevant issue that we had not previously described clearly. We utilized Actin1 of actin gene family and UBQ10 of poly-ubiquitin gene family as the reference genes in our study. We have also analyzed that the expression of Actin1 and UBQ10 is consistent in salt stress. We added a sentence in the revision to address this issue: “All of the quantified proteins were normalized with protein intensity of Actin1 and UBQ10”

Point 5: Here it is not clear after how many days germination seedling's pictures were collected to test germination rate.

Response 5: We have tested the germination rate of those seedlings at 8 days after sowing (DAS) and took the pictures of seedlings at the same day. We have also described this process in the figure legend of Figure 1A ,1B: “Col4, LUC-vector, hy2 mutant and LUC-HY2 overexpression lines were sown respectively on 1/2 MS medium (as mock) or 1/2 MS medium containing indicated concentration of 200 mM NaCl for 4 d.” “Seeds were transferred to 1/2 MS containing 200 mM NaCl, and then the seed germination rates were calculated at 3-5 d.”

  1. Yang, Y.; Guo, Y. Unraveling salt stress signaling in plants. Journal of integrative plant biology 2018, 60, 796-804.
  2. Yang, Y.; Guo, Y. Elucidating the molecular mechanisms mediating plant salt-stress responses. The New phytologist 2018, 217, 523-539.
  3. Huh, S.M.; Noh, E.K.; Kim, H.G.; Jeon, B.W.; Bae, K.; Hu, H.C.; Kwak, J.M.; Park, O.K. Arabidopsis annexins AnnAt1 and AnnAt4 interact with each other and regulate drought and salt stress responses. Plant & cell physiology 2010, 51, 1499-1514.

Reviewer 3 Report

The manuscript presented by Piao et al., and his colleagues deals with the role HY2 gene under salt stress during seed germination in Arabidopsis where they investigated that HY2 is a positive regulator of seed germination under salt stress. The manuscript is well organized and written the results obtained by the authors. However, I have several comments on the manuscript below,

  1. First of all, as it is well known from several studies that HY2 participates in the apoplastic and chloroplastic redox signaling networks, being responsible for chlorophyll, do authors have the answer in which pathway HY2 participates to overcome salt stress either which redox states, which hormones and metabolites?
  2. If the Authors have any information what would be the redox amount in wild type and as well as hy2 mutant?
  3. However, the maintaining a balanced of cytosolic Na+/K+ ratio is another key to the salinity tolerance mechanism, so this information is lack to the manuscript either the transporters of Na+/K+ or total Na+/K+ content in the whole seedlings. Is it possible to test the transporter's expression of Na+/K+ under salt stress in wild-type and hy2 mutants? The data are not really enough to support the Journal.
  4. Please correct how many reference genes are used to normalize the target genes.
  5. Here it is not clear after how many days germination seedling's pictures were collected to test germination rate.

Author Response

Response to Reviewer 3 Comments

    We thank the reviewer for your interest in our research, and the great efforts in helping improving the quality of our manuscript. We have replied the questions point-by-point as following:

Point 1:The authors well described the main results, which sound with the initial hypothesis, but I think the discussion should be improved, i.e., not limited to the comparison with the current literature or repeating the results.

Response 1: We really appreciated the reviewer’s suggestion, and we carefully revised the discussion : “In this study, we found that HY2 specifically regulated pathways as ion balance, ion transmembrane transport, ROS metabolism, antioxidant response and photosynthesis, etc. Numerous previous studies have shown that salt stress is closely related to antioxidant indexes in plants, such as the content of H2O2, NADH and ATP, as well as Na+/K+ ion balance[1]. Besides, studies have shown that HY2 participates in apoplastic and chloroplastic redox signaling networks, being responsible for chlorophyll biosynthesis[2]. In this study, we found that HY2 regulated the ion osmoregulation, organics regulation and related enzyme activity regulation pathway, which were involved in Na+/K+ homeostasis regulation. Furthermore, a few members of K+ transporter family ATANNs and PRXs family were regulated by HY2 after during salt stress, which directly affect the ion osmoregulation, organics regulation and related enzyme activity regulation to maintain the Na+/K+ homeostasis in salt stress. So we speculated that under salt stress, HY2 would be induced to up regulate the expression, leading to the imbalance of reactive oxygen species and Na+ / K+ in vivo and thus the damage of plants by reactive oxygen species and ion stress, the decrease of chlorophyll content and the serious impact on photosynthesis.”

Point 2: The results are also briefly presented at the end of the introduction, but I think the authors should not anticipate them otherwise the text becomes very redundant.

Response 2: We agree. We carefully revised the introduction part, and deleted the anticipated context that make our manuscript clearer and more definite. Please check the revised introduction for the detail.

Point 3: With respect to the figure, if possible, I suggest avoiding the gene functions to overlap with the bars in the graphs.

Response 3: We thank the reviewer for going through our manuscript rigorously, and we revised the pictures Fig 3C, 3D, Fig 4C, 4D, Fig 5A, 5B, 5C and Fig S3D according to the suggestion given by the reviewer.

Point 4: The manuscript needs moderate English (and typing) revision.

Response 4: We checked our manuscript carefully in the revision, and revised 43 language issue in this version.

  1. Ma, L.; Zhang, H.; Sun, L.; Jiao, Y.; Zhang, G.; Miao, C.; Hao, F. NADPH oxidase AtrbohD and AtrbohF function in ROS-dependent regulation of Na⁺/K⁺homeostasis in Arabidopsis under salt stress. Journal of experimental botany 2012, 63, 305-317.
  2. Sierla, M.; Rahikainen, M.; Salojärvi, J.; Kangasjärvi, J.; Kangasjärvi, S. Apoplastic and chloroplastic redox signaling networks in plant stress responses. Antioxidants & redox signaling 2013, 18, 2220-2239.

Round 2

Reviewer 3 Report

The present manuscripts by Piao et al on the role of the HY2 gene to regulate NaCl signaling during seed germination in Arabidopsis. The authors edited and corrected the manuscript considering the reviewer's directions and suggestions. More importantly, they answered the issues of authors and added some text for further clarifications. In my opinion, the current form of the manuscript could be considered for full publication to the Journal of IJMS.